# Effects of Integrating a Plyometric Training Program During Physical Education Classes on Ballistic Neuromuscular Performance

**DOI:** 10.3390/jfmk10030240

**Published:** 2025-06-25

**Authors:** Diego A. Alonso-Aubin, Ángel Saez-Berlanga, Iván Chulvi-Medrano, Ismael Martínez-Guardado

**Affiliations:** 1Strength Training and Neuromuscular Performance Research Group (STreNgthP), Faculty of Health Sciences—HM Hospitals, University Camilo José Cela, C/Castillo de Alarcón, 49, Villanueva de la Cañada, 28692 Madrid, Spain; diegoalexandre.alonso@ucjc.edu; 2HM Hospitals Health Research Institute, 28015 Madrid, Spain; 3Research Group in Prevention and Health in Exercise and Sport (PHES), Department of Physical Education and Sports, University of Valencia, C/Gascó Oliag, 3, 46010 Valencia, Spain; angel.saez@uv.es (Á.S.-B.); ivan.chulvi@uv.es (I.C.-M.); 4LFE Research Group, Department of Health and Human Performance, Faculty of Physical Activity and Sport Science (INEF), Universidad Politécnica de Madrid, C/Martín Fierro, 7, 28040 Madrid, Spain

**Keywords:** strength, power, youth, force plates

## Abstract

**Objective**: This study examined the effects of a short-duration plyometric training program during physical education on neuromuscular ballistic performance in youth. **Methods**: Thirty-two students were assigned to a control group (CG; *n* = 16; age: 16.76 ± 0.72 years; height: 1.66 ± 0.09 m; body mass: 61.38 ± 6.07 kg) or an experimental group (EG; *n* = 16; age: 16.56 ± 0.62 years; height: 1.69 ± 0.09 m; body mass: 61.90 ± 7.83 kg). Both groups completed pre- and post-intervention Countermovement Jump (CMJ) tests using force plates. Over a four-week period, the EG completed eight sessions. Both the EG and the CG participated in 40 min sessions incorporating speed games, directional changes, and agility exercises. Paired *t*-tests and Cohen’s d were used for analysis. **Results**: The EG showed significant improvements in jump height (*p* = 0.006, ES = 0.83), jump momentum (*p* = 0.008, ES = 0.80), and take-off velocity (*p* = 0.003, ES = 0.93), with a decrease in peak propulsive power (*p* = 0.01, ES = 0.77). In contrast, the CG exhibited declines in multiple metrics, including jump height, jump momentum, and take-off velocity. **Conclusions**: These findings suggest that integrating plyometric training into physical education classes can effectively enhance neuromuscular performance in youth. Implementing structured training protocols within school programs may optimize strength, power, and movement efficiency, benefiting long-term athletic development.

## 1. Introduction

Plyometric training has been a popular method for improving athletic performance, particularly in sports that require explosive power and vertical ability [1]. This training incorporates a stretch–shortening cycle (SSC), which involves a rapid eccentric muscle contraction followed by an immediate concentric contraction [2].

The advantage of plyometric exercises is that they optimize muscle–tendon behavior throughout the entire range of motion in each repetition, providing greater training stimuli [3]. The scientific literature reports several benefits of plyometric training, including a reduced risk of injuries, enhanced neuromuscular function, and improved sports performance [4,5,6]. A recent study by Ricart-Luna et al. [7] evaluated a four-week mesocycle in highly trained adolescent players from the Valencia Basket Club. Three weekly 45 min sessions significantly improved lower-limb power (*p* < 0.05), with gains in the Countermovement Jump (CMJ) and Broad Jump tests. Agility also improved, as shown by better Compass Drill performances. These results highlight the value of plyometric training for enhancing jumping ability and movement speed in high-performance sports, supporting its integration into youth training programs.

Physical education classes could provide an excellent environment for achieving some of these benefits through plyometric training, but this topic remains underexplored in this context, partly because the official curriculum presents general and non-specific content, leading each teacher to adopt different methodologies based on their own interests. Nonetheless, several studies have demonstrated significant improvements in strength, balance, and speed through the application of strength-oriented programs at the onset of physical education sessions [8,9]. Marzouki et al. explored how different surfaces could improve physical fitness in youth, reporting significant improvements in sprinting, jumping, and the speed of direction changes [10,11]. Another study explored biomechanical parameters, finding a positive and notable effect on muscle strength, balance, and flexibility in primary-school students [12]. However, the specific mechanisms underlying these improvements, as well as the optimal training parameters, are still being investigated.

The use of force plates has become increasingly common in research for assessing neuromuscular performance and for monitoring the effects of plyometric training, as they are able to measure the ground-reaction forces generated during various jumping and landing activities [13]. By recording the vertical ground-reaction forces during movement as CMJs, researchers can obtain metrics such as jump height, peak force, and rate of force development, providing important insights into neuromuscular adaptations as well as injury risk information [14].

Due to the growing interest in the benefits and potential advantages of plyometric training in young individuals, as well as the scarcity of studies evaluating plyometric adaptations, this study aims to examine the impact of a short-duration plyometric training program, conducted during physical education classes, on neuromuscular ballistic performance in youth using the CMJ test.

Moreover, there is currently no study assessing the effect of plyometric training on CMJ performance and neuromuscular ballistic performance conducted during physical education classes using force plates.

## 2. Materials and Methods

This study used a longitudinal experimental design to assess the neuromuscular ballistic capabilities of youth through force–time metrics measured during a vertical force-plate CMJ test. Participants followed a plyometric training program integrated into their physical education classes.

### 2.1. Participants

Thirty-two youth participated in the study. Participants were recruited from a Spanish college if they met the following criteria: (1) no participation in other physical or training programs during the intervention and (2) ability to complete all the scheduled sessions, including session assessments. The exclusion criteria were the following: (1) having had any musculoskeletal injury in the last three months; (2) having any medical condition (physical or mental) that would prevent participation in the study; and (3) having consumed any type of energy drink or stimulant prior to the neuromuscular assessments. The participants were also asked not to change their sports habits during the intervention, so they maintained their regular extracurricular sports practices.

This study was approved by the institutional review board of the Camilo José Cela University (16_23_RNM_FP), and all participants provided written informed consent from their parents, and they signed an assent consent form before participation. This study followed the principles of the Declaration of Helsinki of the World Medical Association.

A priori sample size estimation was conducted using G*Power (version 3.1.9.7), based on an independent samples *t*-test. Assuming a large effect size (Cohen’s d = 1.35), a significance level of α = 0.05, and a statistical power of 0.95, the minimum required sample size was calculated to be 32 participants (16 per group). This requirement was met, ensuring sufficient power to detect large effects of the 4-week plyometric program.

Participants were randomly assigned into two groups using an electronic procedure (https://www.randomizer.org, accessed on 15 March 2024): a control group (CG) (*n*: 16; age: 16.76 ± 0.72 years; height: 1.66 ± 0.09 m; body mass: 61.38 ± 6.07 kg) and an experimental group (EG) (*n*: 16; age: 16.56 ± 0.62 years; height: 1.69 ± 0.09 m; body mass: 61.90 ± 7.83 kg). Since these were regular physical education sessions, attendance at the sessions was monitored, and it was found that all participants maintained an adherence of at least 85%.

### 2.2. Procedures

Prior to the first assessments, the students underwent familiarization with the CMJ test during the preceding week. This familiarization process took place over two sessions, with a total of six jumps. The study lasted for six weeks, with the first and last weeks dedicated to neuromuscular performance assessments, while the other four weeks were allocated to the implementation of the plyometric training program. Two lead investigators, both certified with the CSCS and with over 10 years of experience as researchers and physical education teachers, conducted the assessments as well as the plyometric training program. The chronological outline of the study can be found in Figure 1.

### 2.3. Assessments

The vertical ground-reaction force applied during each jumping test was collected using a wireless dual-force-plate system with a sample rate of 1000 Hz (Hawkin Dynamics, 4th Generation; Westbrook, ME, USA) and Hawkin Dynamics software (version 9.6.0). The system was zeroed before each trial. The force–time and force–velocity curves of a CMJ obtained through the Hawkin Dynamics software (v9.6.1) can be viewed in Figure 2.

For the CMJ test, the participants were instructed to perform as high a jump as possible, with the following instruction “jump as high and as fast as possible”. Participants had three attempts, each separated by a 30 s rest period, in both the pre- and post-intervention tests. First, participants stepped onto the platforms and remained still and upright, with their knees, ankles, and hips extended. Additionally, their hands had to be placed in an “akimbo” position on their hips before each jump. After each jump assessment, the average of the three jumps was calculated for analysis.

#### Plyometric Training Program

Both the CG and the EG underwent 40 min sessions based on speed games, changes of direction, and agility. Both groups had similar physical education and recreational sport schedules, as the students belonged to the same public school and had the same teacher.

The students assigned to the EG participated in a plyometric training program for four weeks, twice a week, totaling eight sessions. The plyometric training was supervised by an experienced professional in strength and plyometric training for young individuals. Before performing the plyometric training, participants completed a general warm-up consisting of a 2 min run and general joint mobility exercises. A 30 s rest period was provided after each set of every exercise, with a 60 s rest between exercises to allow for adequate recovery to ensure maximum execution intensity. The plyometric training program is shown in Table 1.

### 2.4. Statistical Analysis

A descriptive analysis of the neuromuscular performance, pre- and post-intervention, in CMJ force–time metrics was performed, including the mean, standard deviation, and 95% confidence interval for both the CG and the EG. The Shapiro–Wilk normality test revealed that all the metrics were normally distributed (*p* ≥ 0.05). To compare the pre- and post-intervention force–time metrics within each group, a paired *t*-test was conducted. To explore the differences between the EG and the CG in neuromuscular performance variables, an independent samples *t*-test was performed. The effect size was calculated using Cohen’s d with the following scale: trivial (<0.2), small (0.2–0.5), moderate (0.5–0.8), large (0.8–1.3), and very large (>1.3). Statistical significance was set at *p* ≤ 0.05. All statistical analyses were performed using IBM SPSS Statistics for Windows (v30.0.0), and figures were prepared using JASP software (JASP Team, version 0.17.3; Amsterdam, The Netherlands) and the Mind the Graph platform.

## 3. Results

All the students met the inclusion criteria and completed the plyometric training program, and no injuries occurred over the course of the study.

The results of the plyometric training in the EG and CG based on paired *t*-tests can be found in Table 2. For the EG, we found an increase in the jump height (t = 3.24; *p* = 0.006; ES = 0.83), jump momentum (t = 3.10; *p* = 0.008; ES = 0.80), and take-off velocity (t = 3.67; *p* = 0.003; ES = 0.93) and a decrease in peak propulsive power (t = 2.99; *p* = 0.01; ES = 0.77).

For the CG, we found a decrease in the jump height (t = 2.39; *p* = 0.03; ES = 0.66), jump momentum (t = 2.12; *p* = 0.05; ES = −0.59), time to take-off (t = 3.97; *p* = 0.002; ES = 1.10), take-off velocity (t = 2.25; *p* = 0.04; ES = 0.62), reactive strength index (RSI) (t = −2.95; *p* = 0.01; ES = −0.82), and propulsive impulse (t = 3.37; *p* = 0.006; ES = 0.93).

The results of the plyometric training in the EG and CG based on independent *t*-tests can be found in Table 3. The intergroup analysis revealed statistically significant differences in five neuromuscular performance variables. The EG showed significant increases in the peak propulsive force (t = 2.08; *p* = 0.04; ES = 0.78) and peak relative propulsive force (t = 2.33; *p* = 0.02; ES = 0.88), indicating an improvement in force production capacity during jump take-off. Additionally, the EG exhibited a significant reduction in the take-off time (t = −3.02; *p* = 0.005; ES = −1.14), suggesting an optimization of the concentric phase in the CMJ. Significant differences were found in the RSI (t = 2.54; *p* = 0.01; ES = 0.96). Finally, a significant difference in the relative change in propulsive impulse was observed between the groups (t = −3.29; *p* = 0.003; ES = −1.24), with the CG showing a lesser reduction compared with the EG.

## 4. Discussion

The aim of this study was to examine the impact of a short-duration plyometric training program on neuromuscular ballistic performance. Our results suggest that incorporating a plyometric program into physical education classes can enhance neuromuscular ballistic performance in young individuals.

First, we observed a significant improvement in jump height and jump momentum. These results are in accordance with other studies. For example, one meta-analysis of controlled trials reported that a plyometric training performed over 4 to 16 weeks can be moderately effective in increasing the CMJ height for post-peak heights [15]. Other studies found that a combined plyometric and resistance training program is more effective than plyometric training alone [16]. In this regard, our study has shown that a plyometric training program of just four weeks can increase jump height. However, we also measured other variables, such as jump momentum, which may be a more suitable force–time metric, as it allows us to assess the magnitude of changes in jump height measurements and their relationship with other neuromuscular performance indicators, such as sprint ability [17]. Furthermore, in our case, it was observed that the control group showed a decrease in both jump height and jump momentum, highlighting the relationship between these two metrics and their importance for monitoring training prescription [18].

Jump momentum is the product of the take-off velocity and body mass. We have found an increase in take-off velocity in the EG and a decrease in the CG. This is important as the jump momentum correlates with performance improvements, such as in jump height [19]. Our results align with those reported in other studies, which found a relationship between various factors such as maturation status (with adolescents being more sensitive to changes), training level, and the development of type II muscle fibers [20]. Along these lines, another study reported that there are adaptation windows for plyometric training in the later stages of adolescence, as an increase in leg stiffness may enhance the effectiveness of the SSC [21].

In this sense, a study analyzed the influence of maturation on jump performance in adolescent females aged 12 to 16 after plyometric training, showing improvements in jump capacity as well as in other abilities such as change of direction and sprint performance [1]. These results can also be explained by the execution, in both groups, of exercises based on speed, changes of direction, and agility.

Other important metrics in the analysis of neuromuscular performance are the RSI and mRSI, both of which are influenced by the jump height/flight time and time to take-off or ground contact time and are directly associated with the SSC. The results of our study show an improvement, although not statistically significant, in both the EG and the CG. This may be due to the fact that coaching cues aimed at increasing jump height and speed are sensitive to contact time in adolescents [22]. Along these lines, another study found an improvement in the RSI after low-volume plyometric training in children under 14 years after due an improvement in the SSC [23]. Concerning differences between the groups, our results suggest that the CG experienced a greater improvement in the RSI compared with the EG. Specifically, the RSI in the CG increased by 13.56% (from 0.59 ± 0.12 to 0.67 ± 0.14), whereas the EG showed only a marginal increase of 1.67% (from 0.60 ± 0.11 to 0.61 ± 0.13). The fact that the CG improved more than the EG suggests that the plyometric training program did not fully optimize the expected reactive response, possibly due to cumulative overload or insufficient manipulation of the training parameters. Additionally, the CG may have experienced a spontaneous improvement in the RSI due to the nature of physical education classes, where exposure to games involving directional changes and explosive actions could have promoted greater neuromuscular responsiveness. These findings highlight the importance of optimal plyometric load dosing and planning, ensuring sufficient recovery periods to maximize adaptations in SSC mechanics and the reactive capacity of the participants.

Although the protocol and instructions were the same for both groups, this metric is highly sensitive in this population, so the results should be interpreted with caution. However, it appears that short-term learning effects do not impact force platform assessments in young individuals aged 18 to 22 using the CMJ. Therefore, increased use of the CMJ may enhance the consistency and validity of the measurements [24].

The present study revealed that take-off velocity significantly increased in the EG (t = 3.62; *p* = 0.003; ES = 0.93) compared with the CG, suggesting greater efficiency in the concentric phase of the jump. Additionally, the results indicated that the independent samples *t*-test confirmed significant differences in both the peak propulsive and relative peak propulsive forces. These findings align with previous studies indicating that plyometric training enhances neuromuscular reactivity and SSC efficiency, leading to a faster propulsive phase and a greater capacity to generate force [23].

Several potential explanations account for the aforementioned improvements. First, plyometric training may have optimized tendon stiffness and the reutilization of stored elastic energy, thereby reducing the transition time between the eccentric and concentric phases of the jump [25]. Second, continuous exposure to explosive jumps may have induced neuromuscular adaptations in the synchronization and activation of high-threshold motor units, promoting a more efficient motor response during propulsion [26]. Another possible explanation lies in the optimization of the jump technique within the EG. Participants may have learned to reduce unnecessary compensatory movements and to improve their biomechanical alignment during take-off, enabling them to perform the movement more efficiently [27]. Finally, the reduction in take-off time could also be attributed to a shortened amortization phase, as plyometric training tends to minimize ground contact time to enhance explosiveness in sport movements [28].

Finally, we assessed asymmetries in the force production, but no significant differences were found either before or after the intervention. This is also important, as a well-conducted plyometric training program should not increase the risk of injury, which can be caused by greater differences in force production between both legs.

Despite the results obtained and the novelty of this study, it is important to highlight some limitations. First, the intervention lasted only four weeks, which is a relatively short period, and a longer intervention would have been interesting. This would have allowed us to determine whether the adaptations increase or are maintained in the medium term and to assess whether the control group experiences a decline in neuromuscular performance. Additionally, no programs were conducted in other maturation stages, such as childhood or puberty, to determine whether similar results would be observed in these developmental phases. Furthermore, the sample size, although sufficient to detect large effects, was relatively small. Therefore, further research with larger and more diverse samples is needed to increase the generalizability of the findings. Future research should focus on implementing plyometric training programs across different maturation stages with longer intervention periods.

Including plyometric exercises during the specific warm-up phase of physical education classes is a practical and effective strategy to enhance ballistic neuromuscular performance in students. This type of training, characterized by explosive movements such as jumps, hops, and bounds, stimulates the stretch–shortening cycle of the muscles, promoting improvements in power, speed, and coordination. When incorporated regularly into warm-ups, plyometric work can induce meaningful neuromuscular adaptations without requiring additional class time or significant equipment. Moreover, it prepares the body for subsequent physical tasks, reduces the injury risk, and aligns well with curricular objectives related to motor skill development and athletic ability. Thus, integrating plyometric drills into the warm-up phase not only enhances physical readiness but also contributes to long-term performance improvements in youth.

## 5. Conclusions

Our study highlights that a four-week plyometric training program conducted during physical education classes can improve neuromuscular ballistic performance in adolescents. In this regard, physical education teachers may choose to include well-structured and periodized content of this type of content in their curricula to achieve significant improvements and to monitor the proper development of their students. We recommend including this content during the specific warm-up phase of physical education classes.

## Figures and Tables

**Figure 1 jfmk-10-00240-f001:**
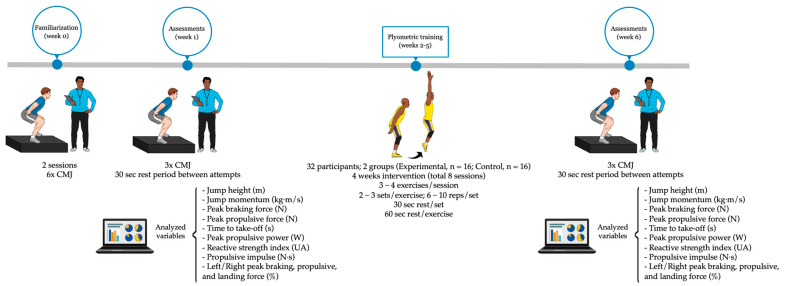
Chronological outline of the study. CMJ, Countermovement Jump. Created with Mindthegraph.com.

**Figure 2 jfmk-10-00240-f002:**
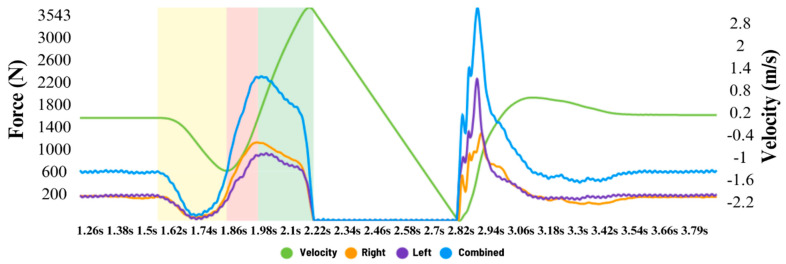
Force-time and velocity-time curves.

**Table 1 jfmk-10-00240-t001:** Plyometric training program. CMJ, Countermovement Jump.

Week 1				
Day	Exercise	Series	Repetitions	Contacts
1	Snap downs	2	8	16
	Pogo jumps	2	10	20
	CMJ to box (0.30 m)	2	6	12
2	Snap downs	3	8	24
	Pogo jumps	3	10	30
	CMJ to box (0.30 m)	3	6	18
Total contacts/week				120
**Week 2**				
**Day**	**Exercise**	**Series**	**Repetitions**	**Contacts**
1	Pogo jumps	3	8	24
	CMJ to box (0.40 m)	3	6	24
	Right unilateral vertical jumps	3	6	18
	Left unilateral vertical jumps	3	6	18
2	Pogo jumps	3	10	30
	CMJ to box (0.40 m)	3	8	24
	Right unilateral vertical jumps	3	8	24
	Left unilateral vertical jumps	3	8	24
Total contacts/week				186
**Week 3**				
**Day**	**Exercise**	**Series**	**Repetitions**	**Contacts**
1	CMJ to box (0.40 m)	3	8	24
	Right unilateral vertical jumps	3	10	30
	Left unilateral vertical jumps	3	10	30
	Drop jump (0.40 m)	3	6	18
2	CMJ to box (0.40 m)	3	8	24
	Right unilateral vertical jumps	3	10	30
	Left unilateral vertical jumps	3	10	30
	Drop jump (0.40 m)	3	8	24
Total contacts/week				210
**Week 4**				
**Day**	**Exercise**	**Series**	**Repetitions**	**Contacts**
1	CMJ to box (0.40 m)	3	8	24
	Right unilateral vertical jumps	3	10	30
	Left unilateral vertical jumps	3	10	30
	Drop jump (0.40 m)	3	6	18
2	CMJ to box (0.40 m)	3	8	24
	Right unilateral vertical jumps	3	10	30
	Left unilateral vertical jumps	3	10	30
	Drop jump (0.40 m)	3	8	24
Total contacts/week				210

**Table 2 jfmk-10-00240-t002:** Results of the plyometric training in the EG and CG (paired-test). RSI, reactive strength index; mRSI, modified reactive strength index; L|R, left–right ratio; SD, standard deviation; CI, confidence interval; t, *t*-test; *p*, significance index; *, *p* ≤ 0.05; **, *p* ≤ 0.01; m, meters; kg·m/s, kilogram-meter per second; N, newton; N/kg, newton per kilogram; s, seconds; m/s, meters per second; W, watt; %, percentage.

Group	Metric	Before Plyometric Training		After Plyometric Training		*t*-test	Significance Index	Effect Size	
		Mean ± SD	CI (95%)	Mean ± SD	CI (95%)	t	*p*	Cohen d	Interpretation
Experimental	Jump height (m)	0.23 ± 0.08	0.14–0.22	0.25 ± 0.09	0.16–0.28	3.24	0.006 **	0.83	Large
	Jump momentum (kg·m/s)	130.58 ± 27.23	94.89–134.98	134.33 ± 29.61	102.76–147.26	3.10	0.008 **	0.80	Large
	Peak braking force (N)	1375.93 ± 302.13	1262.84–1530.49	1354.84 ± 289.40	1353.85–1682.45	0.55	0.59	0.14	Trivial
	Peak relative braking force (N/kg)	227.60 ± 38.64	216.37–257.52	221.19 ± 34.71	232.91–278.20	1.03	0.31	0.26	Small
	Peak propulsive force (N)	1502.40 ± 332.21	1318.28–1627.97	1491.40 ± 337.98	1442.23–1808.53	0.27	0.78	0.07	Trivial
	Peak relative propulsive force (N/kg)	249.00 ± 35.20	227.97–270.52	274.00 ± 43.70	247.49–300.27	0.85	0.40	0.22	Small
	Time to take-off (s)	0.72 ± 0.11	0.50–0.64	0.73 ± 0.12	0.63–0.78	0.76	0.45	0.19	Trivial
	Take-off velocity (m/s)	2.10 ± 0.39	1.67–2.06	2.19 ± 0.44	1.79–2.31	3.62	0.003 **	0.93	Large
	Peak propulsive power (W)	2663.00 ± 733.50	1972.97–2947.71	2549.63 ± 667.40	1918.43–2762.56	2.99	0.01 **	0.77	Moderate
	RSI (UA)	0.60 ± 0.15	0.61–0.74	0.61 ± 0.16	0.54–0.64	1.44	0.17	0.37	Small
	mRSI (UA)	0.33 ± 0.14	0.25–0.41	0.35 ± 0.16	0.26–0.44	1.44	0.17	0.37	Small
	Propulsive impulse (N·s)	277.60 ± 52.08	249.91–306.37	278.14 ± 49.04	251.27–305.48	−0.14	0.88	−0.03	Trivial
	L|R Peak braking force (%)	−3.66 ± 7.61	−2.81–5.55	−2.10 ± 8.46	−2.53–6.23	−1.36	0.19	−0.35	Small
	L|R Peak propulsive force (%)	−1.80 ± 4.26	−2.24–4.58	−1.85 ± 4.17	−2.53–4.84	0.06	0.95	0.01	Trivial
	L|R Peak landing force (%)	−3.14 ± 12.14	−7.25–1.46	1.43 ± 11.10	−10.24–0.73	−1.21	0.24	−0.31	Small
Control	Jump height (m)	0.22 ± 0.09	0.16–0.28	0.18 ± 0.06	0.14–0.22	2.39	0.034 **	0.66	Moderate
	Jump momentum (kg·m/s)	125.01 ± 36.82	102.76–147.26	114.93 ± 33.16	94.89–134.98	2.12	0.05 *	0.58	Moderate
	Peak braking force (N)	1396.66 ± 221.45	1262.28–1530.49	1518.15 ± 271.88	1353.85–1682.45	−1.59	0.13	−0.44	Small
	Peak relative braking force (N/kg)	236.94 ± 34.04	216.37–257.52	255.55 ± 37.47	232.91–278.20	−1.49	0.16	−0.41	Small
	Peak propulsive force (N)	1473.12 ± 256.24	1318.28–1627.97	1625.38 ± 303.07	1442.23–1808.53	−2.15	0.05 *	−0.59	Moderate
	Peak relative propulsive force (N/kg)	249.25 ± 35.20	227.97–270.52	273.88 ± 43.66	247.49–300.27	−2.08	0.05 *	−0.57	Moderate
	Time to take-off (s)	0.71 ± 0.13	0.63–0.78	0.57 ± 0.11	0.50–0.64	3.97	0.002 **	1.10	Large
	Take-off velocity (m/s)	2.05 ± 0.43	1.79–2.31	1.86 ± 0.32	1.67–2.06	2.25	0.04 *	0.62	Moderate
	Peak propulsive power (W)	2460.34 ± 806.51	1972.97–2947.71	2340.50 ± 698.44	1918.43–2762.56	1.09	0.29	0.30	Small
	RSI (UA)	0.59 ± 0.08	0.54–0.64	0.67 ± 0.10	0.61–0.74	−2.95	0.01 *	−0.82	Large
	mRSI (UA)	0.31 ± 0.09	0.25–0.36	0.32 ± 0.08	0.26–0.37	−0.46	0.65	−0.12	Trivial
	Propulsive impulse (N·s)	261.12 ± 69.81	218.93–303.31	233.30 ± 69.68	191.19–275.41	3.37	0.006 **	0.93	Large
	L|R Peak braking force (%)	1.37 ± 6.92	−2.81–5.55	1.85 ± 7.25	−2.54–6.23	−0.59	0.56	−0.16	Trivial
	L|R Peak propulsive force (%)	1.17 ± 5.64	−2.24–4.58	1.24 ± 5.96	−2.35–4.84	−0.11	0.91	−0.03	Trivial
	L|R Peak landing force (%)	−2.89 ± 7.21	−7.52–1.46	−5.48 ± 7.86	−10.24–0.73	1.23	0.23	0.34	Trivial

**Table 3 jfmk-10-00240-t003:** Results of the plyometric training in the EG and CG (independent *t*-test). Note: H_a_ μ_1_ ≠ μ_2._ RSI, reactive strength index; mRSI, modified reactive strength index; L|R, left–right ratio; t, *t*-test; df, degrees of freedom; *p*, significance index; *, *p* ≤ 0.05; **, *p* ≤ 0.01; SE, standard error; CI, confidence interval; m, meters; kg·m/s, kilogram-meter per second; N, newton; N/kg, newton per kilogram; s, seconds; m/s, meters per second; W, watt; %, percentage.

Metric	*t*-test (EG–CG)	Significance Index	Mean Difference	SE of the Difference	CI (95%)		Effect Size	
	t	*p*			Lower	Upper	Cohen d	Interpretation
System weight	−0.88	0.38	−4.06	4.59	−13.51	5.37	−0.33	Small
Jump height (m)	1.12	0.27	−0.01	0.01	−0.05	0.01	−0.42	Small
Jump momentum (kg·m/s)	−1.38	0.17	−6.33	4.58	−15.76	3.09	−0.52	Moderate
Peak braking force (N)	1.74	0.09	142.57	81.69	−25.35	310.50	0.66	Moderate
Peak relative braking force (N/kg)	1.87	0.07	25.02	13.31	−2.34	52.39	0.71	Moderate
Peak propulsive force (N)	2.08	0.04 *	163.25	78.48	1.92	324.59	0.78	Moderate
Peak relative propulsive force (N/kg)	2.33	0.02 *	29.94	12.85	3.53	56.36	0.88	Large
Time to take-off (s)	−3.02	0.005 **	−0.11	0.03	−0.19	−0.03	−1.14	Large
Take-off velocity (m/s)	−1.22	0.23	−0.09	0.08	−0.26	0.06	−0.46	Small
Peak propulsive power (W)	−0.05	0.95	−6.47	109.63	−231.83	218.89	−0.02	Trivial
RSI (UA)	2.53	0.01 *	0.09	0.03	0.01	0.17	0.96	Large
mRSI (UA)	1.27	0.21	0.03	0.02	−0.02	0.08	0.48	Small
Propulsive impulse (N·s)	−3.28	0.003 **	−28.35	8.63	−46.10	−10.61	−1.24	Large
L|R Peak braking force (%)	−0.74	0.46	−1.07	1.43	−4.02	1.88	−0.28	Small
L|R Peak propulsive force (%)	0.11	0.90	0.12	1.03	−2.01	2.25	0.04	Trivial
L|R Peak landing force (%)	−1.59	0.12	−7.16	4.48	−16.39	2.05	−0.60	Moderate
L|R Braking impulse index	−0.16	0.87	−0.57	3.55	−7.89	6.73	−0.06	Trivial
L|R Propulsive impulse index	−0.02	0.98	−0.03	1.51	−3.14	3.07	0.00	Trivial
L|R Landing impulse index	0.89	0.37	8.59	9.55	−11.03	28.22	0.34	Small

## Data Availability

The data that support the findings of this study are available on request from the corresponding author. The data are not publicly available due to privacy or ethical restrictions.

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
