# Peer review of "Effects of Integrating a Plyometric Training Program During Physical Education Classes on Ballistic Neuromuscular Performance"

_jfmk, 2025, doi:10.3390/jfmk10030240_

Round 1
Reviewer 1 Report
Comments and Suggestions for Authors
Dear researchers,
Congratulations for the work, I send my comments or suggestions to the paper:
Introduction
Lines 49 to 56. In this paragraph you point out that the Physical Education class is a space for this type of training or exercises for the improvements you are looking for. The two articles they cite effectively mention an intervention in school children and in the Physical Education class. However, and considering the ministerial curricular regulations, how does the content of the physical education class relate to the expected objective, I ask this question because one of the studies they mention indicated that they participated in a normal way in the Physical Education class, while the other quote points out that this type of training could be a substitute for some exercises in the class.
Methods
Are the objective of this section and the introduction the same?
What is the age of the participants, are there differences between men and women?
Figure 1 is very illustrative, however, the central figure 3 and its description say, for example - 3 - 4 exercise/session I think it gives the impression that it is a negative number, if it is not, you should make use of a dot or other sign.
They indicate that they made three jumps, is there a protocol?
Tables 1 is missing separation line
According to your title, it is not clear when this training was applied. Please clarify if it was before, in class or after class, it is not entirely clear.
Results
Table 2 is missing a separator line
Discussion
Here they indicate another objective.
I believe that the strengths are important, as well as practical implications and future lines of research.
Conclusions
I think it is important to clarify the moment of physical education class. Another important point is that they pointed out that this work can be used by teachers, but what curricular adaptations for the development of physical education content do they suggest? I believe that this point should be included in future lines of research or practical applications.
Author Response
Dear researchers,
Congratulations for the work, I send my comments or suggestions to the paper:
Dear reviewer,
Thank you very much for your kind words and for taking the time to provide such valuable feedback. In the following lines, you will find the adjustments made in accordance with your suggestions.
Best regards.
Introduction
Lines 49 to 56. In this paragraph you point out that the Physical Education class is a space for this type of training or exercises for the improvements you are looking for. The two articles they cite effectively mention an intervention in school children and in the Physical Education class. However, and considering the ministerial curricular regulations, how does the content of the physical education class relate to the expected objective, I ask this question because one of the studies they mention indicated that they participated in a normal way in the Physical Education class, while the other quote points out that this type of training could be a substitute for some exercises in the class.
In Spain, within the context of secondary education, there is an education law that establishes the contents to be taught at each grade level. Specifically, for these groups, the curriculum includes content related to jumps or jumping events, athletic skills, and plyometric training.
However, specific protocols have not been studied in this context, as the contents are general and non-specific, which means each teacher adopts a different methodology based on their own interests.
We have changed the paragraph to clarify:
“Physical education classes could provide an excellent environment to achieve some of these benefits through plyometric training, but this topic remains underexplored in this context, partly because the official curriculum presents general and non-specific content, leading each teacher to adopt different methodologies based on their own interests”.
Methods
Are the objective of this section and the introduction the same? Yes, we have changed to “neuromuscular ballistic performance in youth using the CMJ test.”
What is the age of the participants, are there differences between men and women? Ages are presented in lines 94 and 95. There are not differences between men and women.
Figure 1 is very illustrative, however, the central figure 3 and its description say, for example - 3 - 4 exercise/session I think it gives the impression that it is a negative number, if it is not, you should make use of a dot or other sign. Changed
They indicate that they made three jumps, is there a protocol?
Yes, in the scientific literature, it is common to report the performance of three jumps to ensure good reliability and validity of the measurements.
Warr, D. M., Pablos, C., Sánchez-Alarcos, J. V., Torres, V., Izquierdo, J. M., & Carlos Redondo, J. (2020). Reliability of measurements during countermovement jump assessments: Analysis of performance across subphases. Cogent Social Sciences, 6(1). https://doi.org/10.1080/23311886.2020.1843835
Fahey, Jack & McMahon, John & Ripley, Nicholas. (2024). Test Re-Test Reliability of Countermovement Jump, Single Leg Countermovement Jump, and Countermovement Rebound Jump Force Plate Metrics in Female Football Players: Vertical Jump Test-Rest Reliability in Female Youth Football Players. International Journal of Strength and Conditioning. 4. 10.47206/ijsc.v4i1.338.
Tables 1 is missing separation line
Fixed, thank you.
According to your title, it is not clear when this training was applied. Please clarify if it was before, in class or after class, it is not entirely clear.
Changed to “Effects of a Plyometric Training Program During Physical Education Classes on Ballistic Neuromuscular Performance”
Results
Table 2 is missing a separator line
We believe this is due to the page break. We will inform the editorial team to adjust it according to their style guide.
Discussion
Here they indicate another objective. Yes, we have fixed.
I believe that the strengths are important, as well as practical implications and future lines of research.
Conclusions
I think it is important to clarify the moment of physical education class. Another important point is that they pointed out that this work can be used by teachers, but what curricular adaptations for the development of physical education content do they suggest? I believe that this point should be included in future lines of research or practical applications.
We have changed the title, as we believe you are right with your suggestion.
We added this paragraph in the discussion: “Including plyometric exercises during the specific warm-up phase of physical education classes is a practical and effective strategy to enhance ballistic neuromuscular performance in students. This type of training, characterized by explosive movements such as jumps, hops, and bounds, stimulates the stretch-shortening cycle of the muscles, promoting improvements in power, speed, and coordination. When incorporated regularly into warm-ups, plyometric work can induce meaningful neuromuscular adaptations without requiring additional class time or significant equipment. Moreover, it prepares the body for subsequent physical tasks, reduces injury risk, and aligns well with curricular objectives related to motor skill development and athletic ability. Thus, integrating plyometric drills into the warm-up not only enhances physical readiness but also contributes to long-term performance improvements in youth.”
We added this sentence to the conclusions: “We recommend including this content during the specific warm-up phase of physical education classes.”
Reviewer 2 Report
Comments and Suggestions for Authors
First of all, I would like to thank the editors for allowing me to review this article. I would also like to congratulate the authors for their work.
Some recommendations for improvement:
Abstract
- Include the average age of the student body.
- What did the CG do during those 8 weeks?
- Include the duration of the sessions.
Introduction
- Do not include statistical values in the introduction, by saying that the results of those studies were significant it is understood that they were p<0.05.
- I find this a very interesting approach, but I need you to explain why this is interesting to include in the physical education class. We agree that it has benefits for sports performance, but is this interesting for all students? Is there any other reason to include plyometric training during physical education class, instead of working on other contents?
Materials and methods
- What type of sampling was used?
- Does your inclusion criterion 1 mean that they did not participate in any other type of training, or that they did not start a new training protocol while in the intervention? I refer to this because adolescents tend to engage in sporting activities, so it would be odd if none engaged in any type of physical activity. So, were all those who played some sport excluded, or those who started a new one during the intervention?
- Were participants who missed a single session excluded, or was some leeway allowed for attendance? For example, were those who missed more than 10 or 20% of the program excluded.
- Line 84: change “The” to lowercase “the”.
- In the participants section, was the minimum sample size calculated to conduct the research?
- Was it the researchers' decision to leave 30 seconds between each repetition, or was it based on previous research? Is it enough time?
- Information on the session itself is missing. Was there a specific warm-up in the sessions adapted to this type of activity?
- Were the same researchers in charge of carrying out the protocol and measuring the adolescents in the pre- and post-test?
Results
- Line 148: met the inclusion criteria.
- Include the same number of decimal places in all p-values.
- In Table 3 it is difficult to understand whether it is the EG value minus the CG value or vice versa. Specify this in the header of the table itself (where it says t-test).
Discussion
- Line 178: remove the word “authors”.
- Do not include statistical values in the discussion.
- Include a paragraph on practical applications or future lines of research.
Author Response
First of all, I would like to thank the editors for allowing me to review this article. I would also like to congratulate the authors for their work.
Dear reviewer,
Thank you very much for your kind words and for taking the time to provide such valuable feedback. In the following lines, you will find the adjustments made in accordance with your suggestions.
Best regards.
Some recommendations for improvement:
Abstract
- Include the average age of the student body. Included.
Thirty-two students were assigned to a control group (CG, n=16; age: 16.76±0.72 years; height: 1.66±0.09m; body mass: 61.38±6.07kg) or an experimental group (EG, n=16; age: 16.56±0.62 years; height: 1.69±0.09m; body mass: 61.90±7.83kg).
- What did the CG do during those 8 weeks? Fixed.
- Include the duration of the sessions. Included. “Over a four-week period, the EG completed eight sessions. Both the EG and the CG participated in 40-minute sessions incorporating speed games, directional changes, and agility exercises. Paired t-tests and Cohen’s d were used for analysis”
Introduction
- Do not include statistical values in the introduction, by saying that the results of those studies were significant it is understood that they were p<0.05.
Fixed.
- I find this a very interesting approach, but I need you to explain why this is interesting to include in the physical education class. We agree that it has benefits for sports performance, but is this interesting for all students? Is there any other reason to include plyometric training during physical education class, instead of working on other contents?
We added this paragraph to clarify “Physical education classes could provide an excellent environment to achieve some of these benefits through plyometric training, but this topic remains underexplored in this context, partly because the official curriculum presents general and non-specific content, leading each teacher to adopt different methodologies based on their own interests. Nonetheless, several studies have demonstrated significant improvements in strength, balance and speed through the application of strength-oriented programs at the onset of Physical Education sessions [8,9].”
Materials and methods
- What type of sampling was used?
Convenience sampling.
- Does your inclusion criterion 1 mean that they did not participate in any other type of training, or that they did not start a new training protocol while in the intervention? I refer to this because adolescents tend to engage in sporting activities, so it would be odd if none engaged in any type of physical activity. So, were all those who played some sport excluded, or those who started a new one during the intervention?
Participants who started a new training program during the intervention were excluded. As stated in exclusion criterion 4, participants were asked not to change their sports habits during the intervention period, thus maintaining their regular extracurricular sports practices
- Were participants who missed a single session excluded, or was some leeway allowed for attendance? For example, were those who missed more than 10 or 20% of the program excluded.
(Line 102-103). Given that it was a short-term program, participants were required to attend 85% of the sessions.
- Line 84: change “The” to lowercase “the”.
Changed.
- In the participants section, was the minimum sample size calculated to conduct the research?
We added in line 98 to 102: “A priori sample size estimation was conducted using G*Power (version 3.1.9.7), based on an independent samples t-test. Assuming a large effect size (Cohen’s d = 1.35), a significance level of α = 0.05, and a statistical power of 0.95, the minimum required sample size was calculated to be 32 participants (16 per group). This requirement was met, ensuring sufficient power to detect large effects of the 4-week plyometric program.
We added in line 278 to 280: “Furthermore, the sample size, although sufficient to detect large effects, was relatively small. Therefore, further research with larger and more diverse samples is needed to increase the generalizability of the findings”-
- Was it the researchers' decision to leave 30 seconds between each repetition, or was it based on previous research? Is it enough time?
Based on previous studies on reliability and validity, as well as personal experience using the CMJ test in other research, the 30-second interval between repetitions is the most commonly used and is sufficiently long to allow for valid and replicable multiple attempts.
Warr, D. M., Pablos, C., Sánchez-Alarcos, J. V., Torres, V., Izquierdo, J. M., & Carlos Redondo, J. (2020). Reliability of measurements during countermovement jump assessments: Analysis of performance across subphases. Cogent Social Sciences, 6(1). https://doi.org/10.1080/23311886.2020.1843835
Fahey, Jack & McMahon, John & Ripley, Nicholas. (2024). Test Re-Test Reliability of Countermovement Jump, Single Leg Countermovement Jump, and Countermovement Rebound Jump Force Plate Metrics in Female Football Players: Vertical Jump Test-Rest Reliability in Female Youth Football Players. International Journal of Strength and Conditioning. 4. 10.47206/ijsc.v4i1.338.
- Information on the session itself is missing. Was there a specific warm-up in the sessions adapted to this type of activity?
Included in lines 144 to 145: Before performing the plyometric training, participants completed a general warm-up consisting of a 2-minute run and general joint mobility exercises.
- Were the same researchers in charge of carrying out the protocol and measuring the adolescents in the pre- and post-test?
Included in lines 114 to 117: Two lead investigators, both certified with the CSCS and with over 10 years of experience as researchers and physical education teachers, conducted the assessments as well as the plyometric training program.
Results
- Line 148: met the inclusion criteria. Fixed.
- Include the same number of decimal places in all p-values. Fixed.
- In Table 3 it is difficult to understand whether it is the EG value minus the CG value or vice versa. Specify this in the header of the table itself (where it says t-test). Fixed.
Discussion
- Line 178: remove the word “authors”. Removed.
- Do not include statistical values in the discussion. Deleted.
- Include a paragraph on practical applications or future lines of research.
We added this paragraph at the end of the discussion section: Including plyometric exercises during the specific warm-up phase of physical education classes is a practical and effective strategy to enhance ballistic neuromuscular performance in students. This type of training, characterized by explosive movements such as jumps, hops, and bounds, stimulates the stretch-shortening cycle of the muscles, promoting improvements in power, speed, and coordination. When incorporated regularly into warm-ups, plyometric work can induce meaningful neuromuscular adaptations without requiring additional class time or significant equipment. Moreover, it prepares the body for subsequent physical tasks, reduces injury risk, and aligns well with curricular objectives related to motor skill development and athletic ability. Thus, integrating plyometric drills into the warm-up not only enhances physical readiness but also contributes to long-term performance improvements in youth.